# Preconditioned Spectral Descent for Deep Learning

**David E. Carlson,**[1] **Edo Collins,**[2] **Ya-Ping Hsieh,**[2] **Lawrence Carin,**[3] **Volkan Cevher**[2]
[1] Department of Statistics, Columbia University
[2] Laboratory for Information and Inference Systems (LIONS), EPFL
[3] Department of Electrical and Computer Engineering, Duke University

## Abstract

Deep learning presents notorious computational challenges. These challenges include, but are not limited to, the non-convexity of learning objectives and estimating the quantities needed for optimization algorithms, such as gradients. While we do not address the non-convexity, we present an optimization solution that exploits the so far unused "geometry" in the objective function in order to best make use of the estimated gradients. Previous work attempted similar goals with preconditioned methods in the Euclidean space, such as L-BFGS, RMSprop, and ADAgrad. In stark contrast, our approach combines a non-Euclidean gradient method with preconditioning. We provide evidence that this combination more accurately captures the geometry of the objective function compared to prior work. We theoretically formalize our arguments and derive novel preconditioned non-Euclidean algorithms. The results are promising in both computational time and quality when applied to Restricted Boltzmann Machines, Feedforward Neural Nets, and Convolutional Neural Nets.

## 1 Introduction

In spite of the many great successes of deep learning, efficient optimization of deep networks remains a challenging open problem due to the complexity of the model calculations, the non-convex nature of the implied objective functions, and their inhomogeneous curvature [6]. It is established both theoretically and empirically that finding a local optimum in many tasks often gives comparable performance to the global optima [4], so the primary goal is to find a local optimum quickly. It is speculated that an increase in computational power and training efficiency will drive performance of deep networks further by utilizing more complicated networks and additional data [14].

Stochastic Gradient Descent (SGD) is the most widespread algorithm of choice for practitioners of machine learning. However, the objective functions typically found in deep learning problems, such as feed-forward neural networks and Restricted Boltzmann Machines (RBMs), have inhomogeneous curvature, rendering SGD ineffective. A common technique for improving efficiency is to use adaptive step-size methods for SGD [25], where each layer in a deep model has an independent step-size. Quasi-Newton methods have shown promising results in networks with sparse penalties [16], and factorized second order approximations have also shown improved performance [18]. A popular alternative to these methods is to use an element-wise adaptive learning rate, which has shown improved performance in ADAgrad [7], ADAdelta [30], and RMSprop [5].

The foundation of all of the above methods lies in the hope that the objective function can be well-approximated by Euclidean (e.g., Frobenius or $\ell_2$) norms. However, recent work demonstrated that the matrix of connection weights in an RBM has a tighter majorization bound on the objective function with respect to the Schatten-$\infty$ norm compared to the Frobenius norm [1]. A majorization-minimization approach with the non-Euclidean majorization bound leads to an algorithm denoted as Stochastic Spectral Descent (SSD), which sped up the learning of RBMs and other probabilistic

models. However, this approach does not directly generalize to other deep models, as it can suffer from loose majorization bounds.

In this paper, we combine recent non-Euclidean gradient methods with element-wise adaptive learning rates, and show their applicability to a variety of models. Specifically, our contributions are:

$i$) We demonstrate that the objective function in feedforward neural nets is naturally bounded by the Schatten-$\infty$ norm. This motivates the application of the SSD algorithm developed in [1], which explicitly treats the matrix parameters with matrix norms as opposed to vector norms.

$ii$) We develop a natural generalization of adaptive methods (ADAgrad, RMSprop) to the non-Euclidean gradient setting that combines adaptive step-size methods with non-Euclidean gradient methods. These algorithms have robust tuning parameters and greatly improve the convergence and the solution quality of SSD algorithm via local adaptation. We denote these new algorithms as RMSspectral and ADAspectral to mark the relationships to Stochastic Spectral Descent and RMSprop and ADAgrad.

$iii$) We develop a fast approximation to our algorithm iterates based on the randomized SVD algorithm [9]. This greatly reduces the per-iteration overhead when using the Schatten-$\infty$ norm.

$iv$) We empirically validate these ideas by applying them to RBMs, deep belief nets, feedforward neural nets, and convolutional neural nets. We demonstrate major speedups on all models, and demonstrate improved fit for the RBM and the deep belief net.

We denote vectors as bold lower-case letters, and matrices as bold upper-case letters. Operations $\odot$ and $\oslash$ denote element-wise multiplication and division, and $\sqrt{\mathbf{X}}$ the element-wise square root of $\mathbf{X}$. $\mathbf{1}$ denotes the matrix with all 1 entries. $||\boldsymbol{x}||_p$ denotes the standard $\ell_p$ norm of $\boldsymbol{x}$. $||\mathbf{X}||_{S^p}$ denotes the Schatten-$p$ norm of $\mathbf{X}$, which is $||\boldsymbol{s}||_p$ with $\boldsymbol{s}$ the singular values of $\mathbf{X}$. $||\mathbf{X}||_{S^\infty}$ is the largest singular value of $\mathbf{X}$, which is also known as the matrix 2-norm or the spectral norm.

## 2 Preconditioned Non-Euclidean Algorithms

We first review non-Euclidean gradient descent algorithms in Section 2.1. Section 2.2 motivates and discusses preconditioned non-Euclidean gradient descent. Dynamic preconditioners are discussed in Section 2.3, and fast approximations are discussed in Section 2.4.

### 2.1 Non-Euclidean Gradient Descent

Unless otherwise mentioned, proofs for this section may be found in [13]. Consider the minimization of a closed proper convex function $F(\boldsymbol{x})$ with Lipschitz gradient $||\nabla F(\boldsymbol{x}) - \nabla F(\boldsymbol{y})||_q \leq L_p||\boldsymbol{x} - \boldsymbol{y}||_p, \forall \boldsymbol{x}, \boldsymbol{y}$, where $p$ and $q$ are dual to each other, and $L_p > 0$ is the smoothness constant. This Lipschitz gradient implies the following majorization bound, which is useful in optimization:

$$F(\boldsymbol{y}) \leq F(\boldsymbol{x}) + \langle \nabla F(\boldsymbol{x}), \boldsymbol{y} - \boldsymbol{x} \rangle + \frac{L_p}{2}||\boldsymbol{y} - \boldsymbol{x}||_p^2. \tag{1}$$

A natural strategy to minimize $F(\boldsymbol{x})$ is to iteratively minimize the right-hand side of (1). Defining the #-operator as $\boldsymbol{s}^\# \in \arg\max_{\boldsymbol{x}} \left\{ \langle \boldsymbol{s}, \boldsymbol{x} \rangle - \frac{1}{2}||\boldsymbol{x}||_p^2 \right\}$, this approach yields the algorithm:

$$\boldsymbol{x}_{k+1} = \boldsymbol{x}_k - \frac{1}{L_p} \left[ \nabla F(\boldsymbol{x}_k) \right]^\# \text{, where k is the iteration count.} \tag{2}$$

For $p = q = 2$, (2) is simply gradient descent, and $\boldsymbol{s}^\# = \boldsymbol{s}$. In general, (2) can be viewed as gradient descent in a non-Euclidean norm.

To explore which norm $||\boldsymbol{x}||_p$ leads to the fastest convergence, we note the convergence rate of (2) is $F(\boldsymbol{x}_k) - F(\boldsymbol{x}^*) = \mathcal{O}(\frac{L_p||\boldsymbol{x}_0 - \boldsymbol{x}^*||_p^2}{k})$, where $\boldsymbol{x}^*$ is a minimizer of $F(\cdot)$. If we have an $L_p$ such that (1) holds and $L_p||\boldsymbol{x}_0 - \boldsymbol{x}^*||_p^2 \ll L_2||\boldsymbol{x}_0 - \boldsymbol{x}^*||_2^2$, then (2) can lead to superior convergence. One such example is presented in [13], where the authors proved that $L_\infty||\boldsymbol{x}_0 - \boldsymbol{x}^*||_\infty^2$ improves a dimension-dependent factor over gradient descent for a class of problems in computer science. Moreover, they showed that the algorithm in (2) demands very little computational overhead for their problems, and hence $|| \cdot ||_\infty$ is favored over $|| \cdot ||_2$.

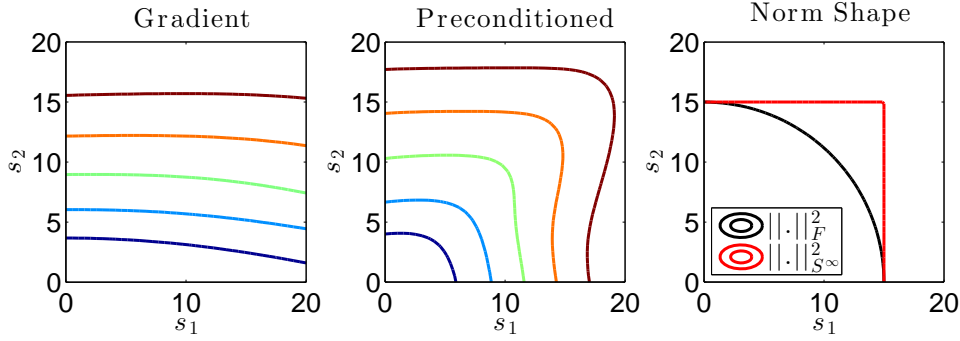

Figure 1: Updates from parameters $\mathbf{W}_k$ for a multivariate logistic regression. (Left) 1st order approximation error at parameter $\mathbf{W}_k + s_1\boldsymbol{u}_1\boldsymbol{v}_1 + s_2\boldsymbol{u}_2\boldsymbol{v}_2$, with $\{\boldsymbol{u}_1, \boldsymbol{u}_2, \boldsymbol{v}_1, \boldsymbol{v}_2\}$ singular vectors of $\nabla_{\mathbf{W}} f(\mathbf{W})$. (Middle) 1st order approximation error at parameter $\mathbf{W}_k + s_1\tilde{\boldsymbol{u}}_1\tilde{\boldsymbol{v}}_1 + s_2\tilde{\boldsymbol{u}}_2\tilde{\boldsymbol{v}}_2$, with $\{\tilde{\boldsymbol{u}}_1, \tilde{\boldsymbol{u}}_2, \tilde{\boldsymbol{v}}_1, \tilde{\boldsymbol{v}}_2\}$ singular vectors of $\sqrt{\mathbf{D}} \odot \nabla_{\mathbf{W}} f(\mathbf{W})$ with $\mathbf{D}$ a preconditioner matrix. (Right) Shape of the error implied by Frobenius norm and the Schatten-$\infty$ norm. After preconditioning, the error surface matches the shape implied by the Schatten-$\infty$ norm and not the Frobenius norm.

As noted in [1], for the log-sum-exp function, $lse(\boldsymbol{\alpha}) = \log \sum_{i=1}^{N} \omega_i \exp(\alpha_i)$, the constant $L_2$ is $\leq 1/2$ and $\Omega(1/\log(N))$ whereas the constant $L_\infty$ is $\leq 1$. If $\boldsymbol{\alpha}$ are (possibly dependent) $N$ zero mean sub-Gaussian random variables, the convergence for the log-sum-exp objective function is improved by at least $\frac{N}{\log^2 N}$ (see Supplemental Section A.1 for details). As well, non-Euclidean gradient descent can be adapted to the stochastic setting [2].

The log-sum-exp function reoccurs frequently in the cost function of deep learning models. Analyzing the majorization bounds that are dependent on the log-sum-exp function with respect to the model parameters in deep learning reveals majorization functions dependent on the Schatten-$\infty$ norm. This was shown previously for the RBM in [1], and we show a general approach in Supplemental Section A.2 and specific results for feed-forward neural nets in Section 3.2. Hence, we propose to optimize these deep networks with the Schatten-$\infty$ norm.

## 2.2 Preconditioned Non-Euclidean Gradient Descent

It has been established that the loss functions of neural networks exhibit pathological curvature [19]: the loss function is essentially flat in some directions, while it is highly curved in others. The regions of high curvature dominate the step-size in gradient descent. A solution to the above problem is to rescale the parameters so that the loss function has similar curvature along all directions. The basis of recent adative methods (ADAgrad, RMSprop) is in preconditioned gradient descent, with iterates

$$\boldsymbol{x}_{k+1} = \boldsymbol{x}_k - \epsilon_k \mathbf{D}_k^{-1} \nabla F(\boldsymbol{x}_k). \tag{3}$$

We restrict without loss of generality the preconditioner $\mathbf{D}_k$ to a positive definite diagonal matrix and $\epsilon_k > 0$ is a chosen step-size. Letting $\langle \boldsymbol{y}, \boldsymbol{x} \rangle_{\mathbf{D}} \triangleq \langle \boldsymbol{y}, \mathbf{D}\boldsymbol{x} \rangle$ and $||\boldsymbol{x}||_{\mathbf{D}}^2 \triangleq \langle \boldsymbol{x}, \boldsymbol{x} \rangle_{\mathbf{D}}$, we note that the iteration in 3 corresponds to the minimizer of

$$\tilde{F}(\boldsymbol{y}) \triangleq F(\boldsymbol{x}_k) + \langle \nabla F(\boldsymbol{x}_k), \boldsymbol{y} - \boldsymbol{x}_k \rangle + \frac{1}{2\epsilon_k} ||\boldsymbol{y} - \boldsymbol{x}_k||_{\mathbf{D}_k}^2. \tag{4}$$

Consequently, for (3) to perform well, $\tilde{F}(\boldsymbol{y})$ has to either be a good approximation or a tight upper bound of $F(\boldsymbol{y})$, the true function value. This is equivalent to saying that the first order approximation error $F(\boldsymbol{y}) - F(\boldsymbol{x}_k) - \langle \nabla F(\boldsymbol{x}_k), \boldsymbol{y} - \boldsymbol{x}_k \rangle$ is better approximated by the *scaled Euclidean norm*. The preconditioner $\mathbf{D}_k$ controls the scaling, and the choice of $\mathbf{D}_k$ depends on the objective function.

As we are motivated to use Schatten-$\infty$ norms for our models, the above reasoning leads us to consider a variable metric non-Euclidean approximation. For a matrix $\mathbf{X}$, let us denote $\mathbf{D}$ to be an element-wise preconditioner. Note that $\mathbf{D}$ is not a diagonal matrix in this case. Because the operations here are element-wise, this would correspond to the case above with a vectorized form of $\mathbf{X}$ and a preconditioner of $\text{diag}(\text{vec}(\mathbf{D}))$. Let $||\mathbf{X}||_{\mathbf{D},S\infty} = ||\sqrt{\mathbf{D}} \odot \mathbf{X}||_{S\infty}$. We consider the following surrogate of $F$,

$$F(\mathbf{Y}) \simeq F(\mathbf{X}_k) + \langle \nabla F(\mathbf{X}_k), \mathbf{Y} - \mathbf{X}_k \rangle + \frac{1}{2\epsilon_k} ||\mathbf{Y} - \mathbf{X}_k||_{\mathbf{D}_k,S\infty}^2. \tag{5}$$

Using the #-operator from Section 2.1, the minimizer of (5) takes the form (see Supplementary Section C for the proof):

$$\mathbf{X}_{k+1} = \mathbf{X}_k - \epsilon_k [\nabla F(\boldsymbol{x}_k) \oslash \sqrt{\mathbf{D}_k}]^{\#} \oslash \sqrt{\mathbf{D}_k}. \tag{6}$$

We note that classification with a softmax link naturally operates on the Schatten-$\infty$ norm. As an illustrative example of the applicability of this norm, we show the first order approximation error for the objective function in this model, where the distribution on the class $y$ depends on covariates $\boldsymbol{x}$, $y \sim$ categorical(softmax($\mathbf{W}\boldsymbol{x}$)). Figure 1 (left) shows the error surfaces on $\mathbf{W}$ without the preconditioner, where the uneven curvature will lead to poor updates. The Jacobi (diagonal of the Hessian) preconditioned error surface is shown in Figure 1 (middle), where the curvature has been made homogeneous. However, the shape of the error does not follow the Euclidean (Frobenius) norm, but instead the geometry from the Schatten-$\infty$ norm shown in Figure 1 (right). Since many deep networks use the softmax and log-sum-exp to define a probability distribution over possible classes, adapting to the the inherent geometry of this function can benefit learning in deeper layers.

### 2.3 Dynamic Learning of the Preconditioner

Our algorithms amount to choosing an $\epsilon_k$ and preconditioner $\mathbf{D}_k$. We propose to use the preconditioner from ADAgrad [7] and RMSprop [5]. These preconditioners are given below:

$$\mathbf{D}_{k+1} = \lambda \mathbf{1} + \sqrt{\mathbf{V}_{k+1}}, \quad \begin{cases} \mathbf{V}_{k+1} = \alpha \mathbf{V}_k + (1-\alpha)\left(\nabla f(\mathbf{X}_k)\right) \odot \left(\nabla f(\mathbf{X}_k)\right), & \text{RMSprop} \\ \mathbf{V}_{k+1} = \mathbf{V}_k + \left(\nabla f(\mathbf{X}_k)\right) \odot \left(\nabla f(\mathbf{X}_k)\right), & \text{ADAgrad} \end{cases}.$$

The $\lambda$ term is a tuning parameter controlling the extremes of the curvature in the preconditioner. The updates in ADAgrad have provably improved regret bound guarantees for convex problems over gradient descent with the iterates in (3) [7]. ADAgrad and ADAdelta [30] have been applied successfully to neural nets. The updates in RMSprop were shown in [5] to approximate the equilibration preconditioner, and have also been successfully applied in autoencoders and supervised neural nets. Both methods require a tuning parameter $\lambda$, and RMSprop also requires a term $\alpha$ that controls historical smoothing.

We propose two novel algorithms that both use the iterate in (6). The first uses the ADAgrad preconditioner which we call ADAspectral. The second uses the RMSprop preconditioner which we call RMSspectral.

### 2.4 The #-Operator and Fast Approximations

Letting $\mathbf{X} = \mathbf{U}\text{diag}(\boldsymbol{s})\mathbf{V}^T$ be the SVD of $\mathbf{X}$, the #-operator for the Schatten-$\infty$ norm (also known as the spectral norm) can be computed as follows [1]: $\mathbf{X}^{\#} = ||\boldsymbol{s}||_1 \mathbf{U}\mathbf{V}^T$.

Depending on the cost of the gradient estimation, this computation may be relatively cheap [1] or quite expensive. In situations where the gradient estimate is relatively cheap, the exact #-operator demands significant overhead. Instead of calculating the full SVD, we utilize a randomize SVD algorithm [9, 22]. For $N \leq M$, this reduces the cost from $\mathcal{O}(MN^2)$ to $\mathcal{O}(MK^2 + MN\log(k))$ with $k$ the number of projections used in the algorithm. Letting $\tilde{\mathbf{U}}\text{diag}(\tilde{\boldsymbol{s}})\tilde{\mathbf{V}}^T \simeq X$ represent the rank-$k+1$ approximate SVD, then the approximate #-operator corresponds to the low-rank approximation and the reweighted remainder, $\mathbf{X}^{\#} \simeq ||\tilde{\boldsymbol{s}}||_1 (\tilde{\mathbf{U}}_{1:k}\tilde{\mathbf{V}}_{1:k} + \tilde{s}_{k+1}^{-1}(\mathbf{X} - \tilde{\mathbf{U}}_{1:K}\text{diag}(\boldsymbol{s}_{1:K})\tilde{\mathbf{V}}_{1:K}^T))$.

We note that the #-operator is also defined for the $\ell_\infty$ norm, however, for notational clarity, we will denote this as $\boldsymbol{x}^{\flat}$ and leave the # notation for the Schatten-$\infty$ case. This $\boldsymbol{x}^{\flat}$ solution was given in [13, 1] as $\boldsymbol{x}^{\flat} = ||\boldsymbol{x}||_1 \times \text{sign}(\boldsymbol{x})$. Pseudocode for these operations is in the Supplementary Materials.

## 3 Applicability of Schatten-$\infty$ Bounds to Models

### 3.1 Restricted Boltzmann Machines (RBM)

RBMs [26, 11] are bipartite Markov Random Field models that form probabilistic generative models over a collection of data. They are useful both as generative models and for "pre-training" deep networks [11, 8]. In the binary case, the observations are binary $\boldsymbol{v} \in \{0,1\}^M$ with connections to latent (hidden) binary units, $\boldsymbol{h} \in \{0,1\}^J$. The probability for each state $\{\boldsymbol{v}, \boldsymbol{h}\}$ is defined

by parameters $\boldsymbol{\theta} = \{\mathbf{W}, \boldsymbol{c}, \boldsymbol{b}\}$ with the energy $-E_{\boldsymbol{\theta}}(\boldsymbol{v}, \boldsymbol{h}) \triangleq \boldsymbol{c}^T \boldsymbol{v} + \boldsymbol{v}^T \mathbf{W} \boldsymbol{h} + \boldsymbol{h}^T \boldsymbol{b}$ and probability $p_{\boldsymbol{\theta}}(\boldsymbol{v}, \boldsymbol{h}) \propto -E_{\boldsymbol{\theta}}(\boldsymbol{v}, \boldsymbol{h})$. The maximum likelihood estimator implies the objective function $\min_{\boldsymbol{\theta}} F(\boldsymbol{\theta}) = -\frac{1}{N} \log \sum_{\boldsymbol{h}} \exp(-E_{\boldsymbol{\theta}}(\boldsymbol{v}_n, \boldsymbol{h})) + \log \sum_{\boldsymbol{v}} \sum_{\boldsymbol{h}} \exp(-E_{\boldsymbol{\theta}}(\boldsymbol{v}_n, \boldsymbol{h}))$.

This objective function is generally intractable, although an accurate but computationally intensive estimator is given via Annealed Importance Sampling (AIS) [21, 24]. The gradient can be comparatively quickly estimated by taking a small number of Gibbs sampling steps in a Monte Carlo Integration scheme (Contrastive Divergence) [12, 28]. Due to the noisy nature of the gradient estimation and the intractable objective function, second order methods and line search methods are inappropriate and SGD has traditionally been used [16]. [1] proposed an upper bound on perturbations to $\mathbf{W}$ of

$$F(\{\mathbf{W} + \mathbf{U}, \boldsymbol{b}, \boldsymbol{c}\}) \leq F(\{\mathbf{W}, \boldsymbol{b}, \boldsymbol{c}\})$$
$$+ \langle \nabla_{\mathbf{W}} F(\{\mathbf{W}, \boldsymbol{b}, \boldsymbol{c}\}), \mathbf{U} \rangle + \frac{MJ}{2} \|\mathbf{U}\|_{S\infty}^2$$

This majorization motivated the Stochastic Spectral Descent (SSD) algorithm, which uses the #-operator in Section 2.4. In addition, bias parameters $\boldsymbol{b}$ and $\boldsymbol{c}$ were bound on the $\ell_\infty$ norm and use the $\flat$ updates from Section 2.4 [1]. In their experiments, this method showed significantly improved performance over competing algorithm for mini-batches of $2J$ and CD-25 (number of Gibbs sweeps), where the computational cost of the #-operator is insignificant. This motivates using the preconditioned spectral descent methods, and we show our proposed RMSspectral method in Algorithm 1.

---

**Algorithm 1** RMSspectral for RBMs

Inputs: $\epsilon_{1,\dots}, \lambda, \alpha, N_b$
Parameters: $\boldsymbol{\theta} = \{\mathbf{W}, \boldsymbol{b}, \boldsymbol{c}\}$
History Terms : $\mathbf{V_W}, \boldsymbol{v_b}, \boldsymbol{v_c}$
**for** $i$=1,... **do**
    Sample a minibatch of size $N_b$
    Estimate gradient $(d\mathbf{W}, d\boldsymbol{b}, d\boldsymbol{c})$
    % Update matrix parameter
    $\mathbf{V_W} = \alpha \mathbf{V_W} + (1 - \alpha) d\mathbf{W} \odot d\mathbf{W}$
    $\mathbf{D_W}^{1/2} = \sqrt{\lambda + \sqrt{\mathbf{V_W}}}$
    $\mathbf{W} = \mathbf{W} - \epsilon_i (d\mathbf{W} \oslash \mathbf{D_W}^{1/2})^\# \oslash \mathbf{D_W}^{1/2}$
    % Update bias term $\boldsymbol{b}$
    $\mathbf{V}_{\boldsymbol{b}} = \alpha \mathbf{V}_{\boldsymbol{b}} + (1 - \alpha) d\boldsymbol{b} \odot d\boldsymbol{b}$
    $\boldsymbol{d}_{\boldsymbol{b}}^{1/2} = \sqrt{\lambda + \sqrt{\boldsymbol{v_b}}}$
    $\boldsymbol{b} = \boldsymbol{b} - \epsilon_i (d\boldsymbol{b} \oslash \boldsymbol{d}_{\boldsymbol{b}}^{1/2})^\flat \oslash \boldsymbol{d}_{\boldsymbol{b}}^{1/2}$
    % Same for $\boldsymbol{c}$
**end for**

---

When the RBM is used to "pre-train" deep models, CD-1 is typically used (1 Gibbs sweep). One such model is the Deep Belief Net, where parameters are effectively learned by repeatedly learning RBM models [11, 24]. In this case, the SVD operation adds significant overhead. Therefore, the fast approximation of Section 2.4 and the adaptive methods result in vast improvements. These enhancements naturally extend to the Deep Belief Net, and results are detailed in Section 4.1.

## 3.2 Supervised Feedforward Neural Nets

Feedforward Neural Nets are widely used models for classification problems. We consider $L$ layers of hidden variables with deterministic nonlinear link functions with a softmax classifier at the final layer. Ignoring bias terms for clarity, an input $\boldsymbol{x}$ is mapped through a linear transformation and a nonlinear link function $\eta(\cdot)$ to give the first layer of hidden nodes, $\boldsymbol{\alpha}_1 = \eta(\mathbf{W}_0 \boldsymbol{x})$. This process continues with $\boldsymbol{\alpha}_\ell = \eta(\mathbf{W}_{\ell-1} \boldsymbol{\alpha}_{\ell-1})$. At the last layer, we set $\boldsymbol{h} = \mathbf{W}_L \boldsymbol{\alpha}_L$ and an $J$-dimensional class vector is drawn $\boldsymbol{y} \sim$ categorical(softmax($\boldsymbol{h}$)). The standard approach for parameter learning is to minimize the objective function that corresponds to the (penalized) maximum likelihood objective function over the parameters $\boldsymbol{\theta} = \{\mathbf{W}_0, \dots, \mathbf{W}_L\}$ and data examples $\{\boldsymbol{x}_1, \dots, \boldsymbol{x}_N\}$, which is given by:

---

**Algorithm 2** RMSspectral for FNN

Inputs: $\epsilon_{1,\dots}, \lambda, \alpha, N_b$
Parameters: $\boldsymbol{\theta} = \{\mathbf{W}_0, \dots, \mathbf{W}_L\}$
History Terms : $\mathbf{V}_0, \dots, \mathbf{V}_L$
**for** $i$=1,... **do**
    Sample a minibatch of size $N_b$
    Estimate gradient by backprop $(d\mathbf{W}_\ell)$
    **for** $\ell = 0, \dots, L$ **do**
        $\mathbf{V}_\ell = \alpha \mathbf{V}_\ell + (1 - \alpha) d\mathbf{W}_\ell \odot d\mathbf{W}_\ell$
        $\mathbf{D}_\ell^{\frac{1}{2}} = \sqrt{\lambda + \sqrt{\mathbf{V}_\ell}}$
        $\mathbf{W}_\ell = \mathbf{W}_\ell - \epsilon_i (d\mathbf{W}_\ell \oslash \mathbf{D}_\ell^{\frac{1}{2}})^\# \oslash \mathbf{D}_\ell^{\frac{1}{2}}$
    **end for**
**end for**

---

$$\boldsymbol{\theta}^{ML} = \arg\min_{\boldsymbol{\theta}} f(\theta) = \frac{1}{N} \sum_{n=1}^{N} \left( -\boldsymbol{y}_n^T \boldsymbol{h}_{n,\boldsymbol{\theta}} + \log \sum_{j=1}^{J} \exp(h_{n,\boldsymbol{\theta},j}) \right) \tag{7}$$

While there have been numerous recent papers detailing different optimization approaches to this objective [7, 6, 5, 16, 19], we are unaware of any approaches that attempt to derive non-Euclidean bounds. As a result, we explore the properties of this objective function. We show the key results here and provide further details on the general framework in Supplemental Section A.2 and the specific derivation in Supplemental Section D. By using properties of the log-sum-exp function

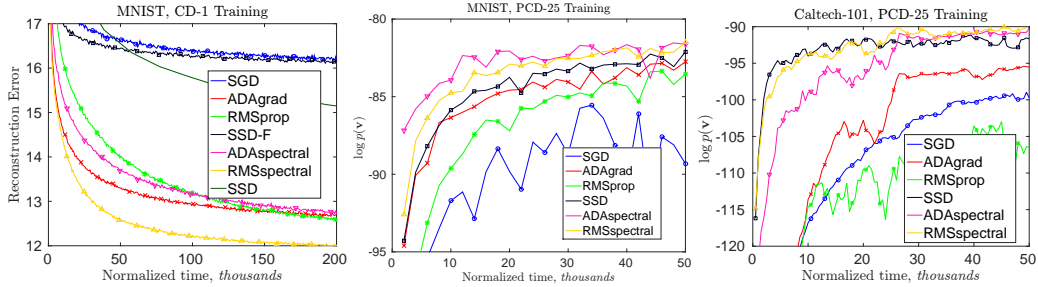

Figure 2: A normalized time unit is 1 SGD iteration (Left) This shows the reconstruction error from training the MNIST dataset using CD-1 (Middle) Log-likelihood of training Caltech-101 Silhouettes using Persistent CD-25 (Right) Log-likelihood of training MNIST using Persistent CD-25

from [1, 2], the objective function from (7) has an upper bound,

$$
\begin{aligned}
f(\boldsymbol{\phi}) \quad \leq \quad & f(\boldsymbol{\theta}) + \langle \nabla_{\boldsymbol{\theta}} f(\boldsymbol{\theta}), \boldsymbol{\phi} - \boldsymbol{\theta} \rangle + \tfrac{1}{N} \sum_{n=1}^{N} \left( \tfrac{1}{2} \max_j (h_{n,\boldsymbol{\phi},j} - h_{n,\boldsymbol{\theta},j})^2 \right. \\
& \left. + 2 \max_j |h_{n,\boldsymbol{\phi},j} - h_{n,\boldsymbol{\theta},j} - \langle \nabla_{\boldsymbol{\theta}} h_{n,\boldsymbol{\theta},j}, \boldsymbol{\phi} - \boldsymbol{\theta} \rangle| \right).
\end{aligned}
\tag{8}
$$

We note that this implicitly requires the link function to have a Lipschitz continuous gradient. Many commonly used links, including logistic, hyperbolic tangent, and smoothed rectified linear units, have Lipschitz continuous gradients, but rectified linear units do not. In this case, we will just proceed with the subgradient. A strict upper bound on these parameters is highly pessimistic, so instead we propose to take a local approximation around the parameter $\mathbf{W}_\ell$ in each layer individually. Considering a perturbation $\mathbf{U}$ around $\mathbf{W}_\ell$, the terms in (8) have the following upper bounds:

$$
(h_{\boldsymbol{\phi},j} - h_{\boldsymbol{\theta},j})^2 \overset{\sim}{\leq} ||\mathbf{U}||_{S^\infty}^2 \, ||\boldsymbol{\alpha}_\ell||_2^2 \, ||\nabla_{\boldsymbol{\alpha}_{\ell+1}} h_j||_2^2 \max_x \, \eta'(x)^2 \, ,
$$

$$
|h_{\boldsymbol{\phi},j} - h_{\boldsymbol{\theta},j} - \langle \nabla_{\boldsymbol{\theta}} h_{\boldsymbol{\theta},j}, \boldsymbol{\phi} - \boldsymbol{\theta} \rangle| \overset{\sim}{\leq} \tfrac{1}{2}||\mathbf{U}||_{S^\infty}^2 ||\boldsymbol{\alpha}_\ell||_2^2 ||\nabla_{\boldsymbol{\alpha}_{\ell+1}} h_j||_\infty ||\nabla_{\boldsymbol{\alpha}_\ell} h_j||_\infty \max_x |\eta''(x)|.
$$

Where $\eta'(x) = \frac{d}{dt}\eta(t)|_{t=x}$ and $\eta''(x) = \frac{d^2}{dt^2}\eta(t)|_{t=x}$. Because both $\boldsymbol{\alpha}_\ell$ and $\nabla_{\boldsymbol{\alpha}_{\ell+1}} h_j$ can easily be calculated during the standard backpropagation procedure for gradient estimation, this can be calculated without significant overhead. Since these equations are bounded on the Schatten-$\infty$ norm, this motivates using the Stochastic Spectral Descent algorithm with the #-operator is applied to the weight matrix for each layer individually.

However, the proposed updates require the calculation of many additional terms; as well, they are pessimistic and do not consider the inhomogenous curvature. Instead of attempting to derive the step-sizes, both RMSspectral and ADAspectral will learn appropriate element-wise step-sizes by using the gradient history. Then, the preconditioned #-operator is applied to the weights from each layer individually. The RMSspectral method for feed-forward neural nets is shown in Algorithm 2.

It is unclear how to use non-Euclidean geometry for convolution layers [14], as the pooling and convolution create alternative geometries. However, the ADAspectral and RMSspectral algorithms can be applied to convolutional neural nets by using the non-Euclidean steps on the dense layers and linear updates from ADAgrad and RMSprop on the convolutional filters. The benefits from the dense layers then propagate down to the convolutional layers.

## 4 Experiments

### 4.1 Restricted Boltzmann Machines

To show the use of the approximate #-operator from Section 2.4 as well as RMSspec and ADAspec, we first perform experiments on the MNIST dataset. The dataset was binarized as in [24]. We detail the algorithmic setting used in these experiments in Supplemental Table 1, which are chosen to match previous literature on the topic. The batch size was chosen to be 1000 data points, which matches [1]. This is larger than is typical in the RBM literature [24, 10], but we found that all algorithms improved their final results with larger batch-sizes due to reduction in sampling noise.

The analysis supporting the SSD algorithm does not directly apply to the CD-1 learning procedure, so it is of interest to examine how well it generalizes to this framework. To examine the effect of CD-1 learning, we used reconstruction error with $J$=500 hidden, latent variables. Reconstruction error is a standard heuristic for analyzing convergence [10], and is defined by taking $||\boldsymbol{v} - \hat{\boldsymbol{v}}||_2$, where $\boldsymbol{v}$ is an observation and $\hat{\boldsymbol{v}}$ is the mean value for a CD-1 pass from that sample. This result is shown in Figure 2 (left), with all algorithms normalized to the amount of time it takes for a single SGD iteration. The full #-operator in the SSD algorithm adds significant overhead to each iteration, so the SSD algorithm does not provide competitive performance in this situation. The SSD-F, ADAspectral, and RMSspectral algorithms use the approximate #-operator. Combining the adaptive nature of RMSprop with non-Euclidean optimization provides dramatically improved performance, seemingly converging faster and to a better optimum.

High CD orders are necessary to fit the ML estimator of an RBM [24]. To this end, we use the Persistent CD method of [28] with 25 Gibbs sweeps per iteration. We show the log-likelihood of the training data as a function of time in Figure 2(middle). The log-likelihood is estimated using AIS with the parameters and code from [24]. There is a clear divide with improved performance from the Schatten-$\infty$ based methods. There is further improved performance by including preconditioners. As well as showing improved training, the test set has an improved log-likelihood of -85.94 for RMSspec and -86.04 for SSD.

For further exploration, we trained a Deep Belief Net with two hidden layers of size 500-2000 to match [24]. We trained the first hidden layer with CD-1 and RMSspectral, and the second layer with PCD-25 and RMSspectral. We used the same model sizes, tuning parameters, and evaluation parameters and code from [24], so the only change is due to the optimization methods. Our estimated lower-bound on the performance of this model is -80.96 on the test set. This compares to -86.22 from [24] and -84.62 for a Deep Boltzmann Machine from [23]; however, we caution that these numbers no longer reflect true performance on the test set due to bias from AIS and repeated overfitting [23]. However, this is a fair comparison because we use the same settings and the evaluation code.

For further evidence, we performed the same maximum-likelihood experiment on the Caltech-101 Silhouettes dataset [17]. This dataset was previously used to demonstrate the effectiveness of an adaptive gradient step-size and Enhanced Gradient method for Restricted Boltzmann Machines [3]. The training curves for the log-likelihood are shown in Figure 2 (right). Here, the methods based on the Schatten-$\infty$ norm give state-of-the-art results in under 1000 iterations, and thoroughly dominate the learning. Furthermore, both ADAspectral and RMSspectral saturate to a higher value on the training set and give improved testing performance. On the test set, the best result from the non-Euclidean methods gives a testing log-likelihood of -106.18 for RMSspectral, and a value of -109.01 for RMSprop. These values all improve over the best reported value from SGD of -114.75 [3].

## 4.2   Standard and Convolutional Neural Networks

Compared to RBMs and other popular machine learning models, standard feed-forward neural nets are cheap to train and evaluate. The following experiments show that even in this case where the computation of the gradient is efficient, our proposed algorithms produce a major speed up in convergence, in spite of the per-iteration cost associated with approximating the SVD of the gradient. We demonstrate this claim using the well-known MNIST and Cifar-10 [15] image datasets.

Both datasets are similar in that they pose a classification task over 10 possible classes. However, CIFAR-10, consisting of 50K RGB images of vehicles and animals, with an additional 10K images reserved for testing, poses a considerably more difficult problem than MNIST, with its 60K greyscale images of hand-written digits, plus 10K test samples. This fact is indicated by the state-of-the-art accuracy on the MNIST test set reaching 99.79% [29], with the same architecture achieving "only" 90.59% accuracy on CIFAR-10.

To obtain the state-of-the-art performance on these datasets, it is necessary to use various types of data pre-processing methods, regularization schemes and data augmentation, all of which have a big impact of model generalization [14]. In our experiments we only employ ZCA whitening on the CIFAR-10 data [15], since these methods are not the focus of this paper. Instead, we focus on the comparative performance of the various algorithms on a variety of models.

We trained neural networks with zero, one and two hidden layers, with various hidden layer sizes, and with both logistic and rectified linear units (ReLU) non-linearities [20]. Algorithm parameters

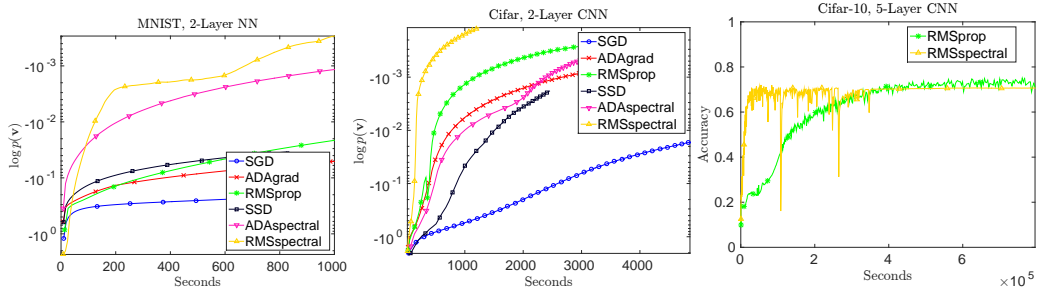

Figure 3: (Left) Log-likelihood of current training batch on the MNIST dataset (Middle) Log-likelihood of the current training batch on CIFAR-10 (Right) Accuracy on the CIFAR-10 test set

can be found in Supplemental Table 2. We observed fairly consistent performance across the various configurations, with spectral methods yielding greatly improved performance over their Euclidean counterparts. Figure 3 shows convergence curves in terms of log-likelihood on the training data as learning proceeds. For both MNIST and CIFAR-10, SSD with estimated Lipschitz steps outperforms SGD. Also clearly visible is the big impact of using local preconditioning to fit the local geometry of the objective, amplified by using the spectral methods.

Spectral methods also improve convergence of convolutional neural nets (CNN). In this setting, we apply the #-operator only to fully connected linear layers. Preconditioning is performed for all layers, i.e., when using RMSspectral for linear layers, the convolutional layers are updated via RM-Sprop. We applied our algorithms to CNNs with one, two and three convolutional layers, followed by two fully-connected layers. Each convolutional layer was followed by max pooling and a ReLU non-linearity. We used $5 \times 5$ filters, ranging from 32 to 64 filters per layer.

We evaluated the MNIST test set using a two-layer convolutional net with 64 kernels. The best generalization performance on the test set after 100 epochs was achieved by both RMSprop and RMSspectral, with an accuracy of 99.15%. RMSspectral obtained this level of accuracy after only 40 epochs, less that half of what RMSprop required.

To further demonstarte the speed up, we trained on CIFAR-10 using a deeper net with three convolutional layers, following the architecture used in [29]. In Figure 3 (Right) the test set accuracy is shown as training proceeds with both RMSprop and RMSspectral. While they eventually achieve similar accuracy rates, RMSspectral reaches that rate four times faster.

## 5 Discussion

In this paper we have demonstrated that many deep models naturally operate with non-Euclidean geometry, and exploiting this gives remarkable improvements in training efficiency, as well as finding improved local optima. Also, by using adaptive methods, algorithms can use the same tuning parameters across different model sizes configurations. We find that in the RBM and DBN, improving the optimization can give dramatic performance improvements on both the training and the test set. For feedforward neural nets, the training efficiency of the propose methods give staggering improvements to the training performance.

While the training performance is drastically better via the non-Euclidean quasi-Newton methods, the performance on the test set is improved for RBMs and DBNs, but not in feedforward neural networks. However, because our proposed algorithms fit the model significantly faster, they can help improve Bayesian optimization schemes [27] to learn appropriate penalization strategies and model configurations. Furthermore, these methods can be adapted to dropout [14] and other recently proposed regularization schemes to help achieve state-of-the-art performance.

**Acknowledgements** The research reported here was funded in part by ARO, DARPA, DOE, NGA and ONR, and in part by the European Commission under grants MIRG-268398 and ERC Future Proof, by the Swiss Science Foundation under grants SNF 200021-146750, SNF CRSII2-147633, and the NCCR Marvel. We thank the reviewers for their helpful comments.

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
