[Supplementary Material]

# Preconditioned Spectral Descent for Deep Learning: Supplemental Material

**David E. Carlson,**[1] **Edo Collins,**[2] **Ya-Ping Hsieh,**[2] **Lawrence Carin,**[3] **Volkan Cevher**[2]
[1] Department of Statistics, Columbia University
[2] Laboratory for Information and Inference Systems (LIONS), EPFL
[3] Department of Electrical and Computer Engineering, Duke University

| Parameter | SGD | ADAgrad | RMSprop | SSD | SSD-F | ADAspec | RMSspec |
|---|---|---|---|---|---|---|---|
| **W** Step size | $10^{-1}$ | $10^{-1}$ | $10^{-3}$ | $1/\sqrt{MJ}$ | $1/\sqrt{MJ}$ | $\frac{1}{2} \times 10^{-3}$ | $\frac{1}{2} \times 10^{-3}$ |
| $\lambda$ | – | $10^{-3}$ | $10^{-3}$ | – | – | $10^{-3}$ | $10^{-3}$ |
| $\alpha$ | – | – | .95 | – | – | – | .95 |
| Projections | – | – | – | – | 30 | 30 | 30 |

Table 1: Parameter Settings for Learning RBMs. RMSprop parameters chosen to match [5]. SGD parameters chosen to match [26]. SSD and A-SSD stepsizes and geometries chosen to match [1]. The stepsize on **W** is given for the RBM. $\lambda$ corresponds to the damping factor in the history terms in the ADA and RMS methods. Projections refers to the numbers of projections used in the Random SVD algorithm [9] in the approximate #-operator (Section 2.4).

| Parameter | SGD | ADAgrad | RMSprop | SSD | ADAspec | RMSspec |
|---|---|---|---|---|---|---|
| Batch size | 100 | 100 | 100 | 500/1500 | 500/1500 | 500/1500 |
| **W** Step size | $5 \cdot 10^{-2}$ | $3 \cdot 10^{-2}$ | $2 \cdot 10^{-2}$ | $2 \cdot 10^{-2}/5 \cdot 10^{-3}$ | $10^{-2}$ | $10^{-3}/10^{-2}$ |
| $\lambda$ | – | $10^{-1}$ | $10^{-1}$ | – | $5 \cdot 10^{-2}$ | $5 \cdot 10^{-2}$ |
| $\alpha$ | – | – | .9 | – | – | .9 |
| Projections | – | – | – | 30 | 30 | 30 |

Table 2: Parameter Settings for Learning NNs/CNNs. See caption of Table 1 for a description of parameters.

---

**Algorithm 3** Sharp Operator

---

Inputs: $\mathbf{X}$
$[\mathbf{U}, \boldsymbol{s}, \mathbf{V}]=\text{SVD}(\mathbf{X})$
Return $\mathbf{X}^{\#} = ||\boldsymbol{s}||_1 \mathbf{U}\mathbf{V}^T$

---

---

**Algorithm 4** Flat Operator

---

Inputs: $\boldsymbol{x}$
Return $\boldsymbol{x}^{\flat} = ||\boldsymbol{x}||_1 \times \text{SIGN}(\boldsymbol{x})$

---

**Algorithm 5** Approximate Sharp Operator

---
Inputs: $\mathbf{X} \in \mathbb{R}^{M \times J}$, approximation level $R$, stabilizing value $\epsilon$
$[\mathbf{U}, \boldsymbol{s}, \mathbf{V}] = \text{RANDOMSVD}(\mathbf{X}, R)$ (Rank $R$ approximation from [9])
$\mathbf{Y} = \mathbf{X} - \mathbf{U}\text{DIAG}(\boldsymbol{s})\mathbf{V}^T$
$l = ||\mathbf{Y}||_{S\infty}$ (power method)
Return $\mathbf{X}^{\#} = ||\boldsymbol{s}||_1 \mathbf{U}\mathbf{V}^T + (||\boldsymbol{s}||_1/(l+\epsilon))\mathbf{Y}$

---

# A Additional Method Details

## A.1 Non-Euclidean Gradient Descent for Deep Learning Models

The purpose of this subsection is to show that $|| \cdot ||_\infty$ and $|| \cdot ||_{S_\infty}$ are much better choices than $|| \cdot ||_2$ and $|| \cdot ||_F$ for analyzing deep learning models. Our insight builds on a novel lemma regarding the properties of the log-sum-exp function. In Section 3 we show that, for many deep learning models, training is equivalent to the optimization problem

$$\min_{\boldsymbol{\theta}} g(\boldsymbol{\theta}) + lse(\boldsymbol{\alpha}(\boldsymbol{\theta})), \tag{7}$$

where $g(\cdot)$ and $\boldsymbol{\alpha}(\cdot)$ are functions depending on the input data and the chosen model, and $lse(\cdot)$ denotes the log-sum-exp function: $lse(\boldsymbol{\alpha}) = \log \sum_{i=1}^{N} \exp(\alpha^i)$. The analysis for $g(\cdot)$ and $\boldsymbol{\alpha}(\cdot)$ must be done on a case-by-case basis. This is discussed for feed-forward neural networks in Section 3.2, and implicitly treated this way for RBMs in [1]. Our aim here is to give a general treatise for $lse(\cdot)$. The following simple theorem sheds some light on the behavior of $lse(\cdot)$, whose proof can be found in the Supplemental Section B.

**Theorem A.1** *Let $F(\cdot) = lse(\cdot)$. Suppose that the entries of $\boldsymbol{\alpha}$ are (possibly dependent) $N$ zero-mean sub-Gaussian random variables. Then it holds that*

$$L_\infty \mathbb{E}||\boldsymbol{\alpha}||_\infty^2 = \mathcal{O}(\log N), \qquad L_2 \mathbb{E}||\boldsymbol{\alpha}||_2^2 = \Omega(\tfrac{N}{\log N}). \tag{8}$$

We remark that Theorem A.1 can be easily strengthened to hold with overwhelming probability; for illustration purposes we will only work with expectation results. As well, the constants $L_\infty$ and $L_2$ are similar, with $L_2 \leq \frac{1}{2}$ and $\Omega(\log N)$ and $L_\infty \leq 1$ [1]. This emphasizes that $\ell_\infty$ is dramatically better for $lse(\cdot)$ optimization. In [1], it was demonstrated that the $\ell_\infty$ bound combined with the properites of $\boldsymbol{\alpha}(\cdot)$ propagates to a $S_\infty$ bound on matrix parameters in an RBM. This similarly happens in the feed-forward neural nets. We give this result in Section 3.2 and give specific mathematical details in Supplemental Section D.

## A.2 Tighter Majorization Bounds in Deep Learning

It is well-known that most deep learning models result in a *non-convex* $g(\cdot)$ and $\alpha(\cdot)$, and one can not reach a global optimum through a convex optimization method. Moreover, obtaining the exact gradient for either $g(\cdot)$ or $lse(\boldsymbol{\alpha}(\cdot))$ is a computationally prohibitive task. Therefore, the goal here is to quickly search for a local minimum of (7) with the help of noisy gradient estimates. Such a procedure is empirically justified, and is what we adopt in this paper.

Our strategy runs as follows. First, we derive a global majorization bound for $g(\boldsymbol{\theta})$, say $g(\boldsymbol{\theta}') \leq g(\boldsymbol{\theta}) + U(\boldsymbol{\theta}, \boldsymbol{\theta}')$. In view of (1), we have arrived at a global majorization bound on the objective function:

$$g(\boldsymbol{\theta}') + lse(\boldsymbol{\alpha}(\boldsymbol{\theta}')) \leq g(\boldsymbol{\theta}) + U(\boldsymbol{\theta}, \boldsymbol{\theta}') + lse(\boldsymbol{\alpha}(\boldsymbol{\theta}))$$
$$+ \langle \nabla lse(\boldsymbol{\alpha}(\boldsymbol{\theta})), \boldsymbol{\alpha}(\boldsymbol{\theta}') - \boldsymbol{\alpha}(\boldsymbol{\theta}) \rangle + \frac{L_p}{2} ||\boldsymbol{\alpha}(\boldsymbol{\theta}') - \boldsymbol{\alpha}(\boldsymbol{\theta})||_p^2 \tag{9}$$
$$\triangleq r(\boldsymbol{\theta}, \boldsymbol{\theta}').$$

Setting $\boldsymbol{\theta}_{k+1} = \arg\min_{\boldsymbol{\theta}'} r(\boldsymbol{\theta}_k, \boldsymbol{\theta}')$ gives our algorithm.

It is evident that our algorithm works well only if the majorization bound (9) is tight. Now, Theorem A.1 implies that using $p = \infty$ improves this majorization bound over $p = 2$ by a factor

of at least $\frac{\log^2 N}{N}$, provided that $\boldsymbol{\alpha}(\boldsymbol{\theta}_k) - \boldsymbol{\alpha}(\boldsymbol{\theta}_{k+1})$ has zero-mean sub-Gaussian entries. (Here, the randomness of $\boldsymbol{\alpha}(\boldsymbol{\theta}_k)$ or $\boldsymbol{\alpha}(\boldsymbol{\theta}_{k+1})$ arises from the noise of gradient estimates.) Because we are changing the parameters we do not necessary expect zero-mean entries, but the scaling result is bound by $|| \cdot ||_\infty$ rather than $|| \cdot ||_2$.

So far, we have shown that $|| \cdot ||_\infty$, instead of the commonly adopted $|| \cdot ||_2$, is a natural choice for analyzing deep learning models. When these bounds are applied and propagated to the parameters by analyzing the max change in $\boldsymbol{\alpha}(\cdot)$, they naturally form a bound on the maximum singular value on the perturbation about matrix parameter, or the Schatten-$\infty$ norm.

## B  Proof of Theorem A.1

We will use the following elementary facts: If $\alpha_1, \alpha_2, ..., \alpha_N$ are (possibly dependent) zero-mean sub-Gaussian random variables, then

$$\left( \mathbb{E} \max_i \alpha_i \right)^2 = \mathcal{O}(\log N), \qquad \left( \mathbb{E} \sum_{i=1}^N \alpha_i^2 \right) = \Theta(N). \tag{10}$$

In [1], it is derived that $L_\infty \leq 1$ for $lse(\cdot)$. Hence it remains to show $L_2 = \Omega(\frac{1}{\log N})$.

Consider the bound

$$lse(\boldsymbol{y}) \leq lse(\boldsymbol{x}) + \langle \nabla lse(\boldsymbol{x}), \boldsymbol{y} - \boldsymbol{x} \rangle + \frac{L_2}{2} ||\boldsymbol{y} - \boldsymbol{x}||_2^2, \tag{11}$$

which must hold for all $\boldsymbol{x}, \boldsymbol{y}$. Let $\boldsymbol{y}$ be the all 0 vector, and let $\boldsymbol{x}$ have $\delta$ in its first entry, and 0 otherwise. Substituting these into (11), we get

$$\log N \leq \log(e^\delta + N - 1) - \frac{\delta e^\delta}{e^\delta + N - 1} + \frac{\delta^2 L_2}{2} \tag{12}$$

where we have used the formula $\nabla lse(x) = \frac{\exp(\boldsymbol{x})}{\sum_{i=1}^N e^{x^i}}$. Setting $\delta = \log N$ and rearranging, we get

$$L_2 \geq 2 \left( \frac{1}{\log^2 N} \log \frac{N}{2N - 1} + \frac{1}{\log N} \cdot \frac{N}{2N - 1} \right) = \Omega \left( \frac{1}{\log N} \right). \tag{13}$$

which is the desired result.

## C  Preconditioned #-Operator Updates

We give the proof of a general form of the iteration (6), which also applies to our $\flat$-operator in the vector case. Let $|| \cdot ||$ be an arbitrary norm and let its corresponding #-operator be $\boldsymbol{s}^\# = \arg\min_{\boldsymbol{x}} \left\{ \langle \boldsymbol{s}, \boldsymbol{x} \rangle - \frac{1}{2} ||\boldsymbol{x}||^2 \right\}$. It is shown in [1] that the minimizer of

$$\min_{\boldsymbol{y}} F(\boldsymbol{x}) + \langle \nabla F(\boldsymbol{x}), \boldsymbol{y} - \boldsymbol{x} \rangle + \frac{1}{2\epsilon} ||\boldsymbol{y} - \boldsymbol{x}||^2$$

is given by

$$\boldsymbol{y} = \boldsymbol{x} - \epsilon [\nabla F(\boldsymbol{x})]^\#. \tag{14}$$

Now, our objective function is given by

$$\min_{\boldsymbol{y}} F(\boldsymbol{x}) + \langle \nabla F(\boldsymbol{x}), \boldsymbol{y} - \boldsymbol{x} \rangle + \frac{1}{2\epsilon} ||\mathbf{D}(\boldsymbol{y} - \boldsymbol{x})||^2 \tag{15}$$

for some positive definite and diagonal $\mathbf{D}$. Notice that $\langle \nabla F(\boldsymbol{x}), \boldsymbol{y} - \boldsymbol{x} \rangle = \langle \mathbf{D}^{-1} \nabla F(\boldsymbol{x}), \mathbf{D}(\boldsymbol{y} - \boldsymbol{x}) \rangle$. Applying (14) to $\boldsymbol{y}' = \mathbf{D}\boldsymbol{y}$ and $\boldsymbol{x}' = \mathbf{D}\boldsymbol{x}$, we see that the minimizer of (15) is $\boldsymbol{y}' = \boldsymbol{x}' - \epsilon[\mathbf{D}^{-1}\nabla F(\boldsymbol{x})]^\#$. Multiplying both sides by $\mathbf{D}^{-1}$ proves (6).

# D Feedforward Neural Net Schatten-$\infty$ Bound Result

Since the softmax classifier log-likelihood objective function has the bound

$$f(\boldsymbol{\phi}) \leq f(\boldsymbol{\theta}) + \langle \nabla_{\boldsymbol{\theta}} f(\boldsymbol{\theta}), \boldsymbol{\phi} - \boldsymbol{\theta} \rangle + \frac{1}{N} \sum_{n=1}^{N} \Big( \frac{1}{2} \max_j (h_{n,\boldsymbol{\phi},j} - h_{n,\boldsymbol{\theta},j})^2$$

$$+ 2 \max_j |h_{n,\boldsymbol{\phi},j} - h_{n,\boldsymbol{\theta},j} - \langle \nabla_{\boldsymbol{\theta}} h_{n,\boldsymbol{\theta},j}, \boldsymbol{\phi} - \boldsymbol{\theta} \rangle| \Big),$$

this requires that we analyze $(h_{n,\boldsymbol{\phi},j} - h_{n,\boldsymbol{\theta},j})^2$ and $|h_{n,\boldsymbol{\phi},j} - h_{n,\boldsymbol{\theta},j} - \langle \nabla_{\boldsymbol{\theta}} h_{n,\boldsymbol{\theta},j}, \boldsymbol{\phi} - \boldsymbol{\theta} \rangle|$. Note that the following relationships hold and are standard

$$\begin{aligned}
||\mathbf{X}\boldsymbol{y}||_2 &\leq ||\mathbf{X}||_{S^\infty} ||\boldsymbol{y}||_2, \\
||\boldsymbol{z}^T \mathbf{X}\boldsymbol{y}||_2 &\leq ||\mathbf{X}||_{S^\infty} ||\boldsymbol{y}||_2 ||\boldsymbol{z}||_2, \\
||\boldsymbol{x} \odot \boldsymbol{y}||_2 &\leq ||\boldsymbol{x}||_\infty ||\boldsymbol{y}||_2.
\end{aligned}$$

Besides, the following elementary formulae hold:

$$\nabla_{\mathbf{W}_\ell} h_j = \big((\nabla_{\boldsymbol{\alpha}_{\ell+1}} h_j) \odot \eta'(\mathbf{W}_\ell \boldsymbol{\alpha}_\ell)\big) \boldsymbol{\alpha}_\ell^T,$$

$$\nabla_{\boldsymbol{\alpha}_\ell} h_j = \mathbf{W}_\ell^T ((\nabla_{\alpha_{\ell+1}} h_j) \odot \eta'(\mathbf{W}_\ell \boldsymbol{\alpha}_\ell)).$$

Considering a single data point, for $(h_{\boldsymbol{\phi},j} - h_{\boldsymbol{\theta},j})^2$, we can get the bound for block-coordinate-wise updates for $\mathbf{W}_\ell$ to $\mathbf{W}_\ell + \mathbf{U}$ by noting

$$\begin{aligned}
&(h_{\mathbf{W}_\ell + \mathbf{U}, j} - h_{\mathbf{W}_\ell, j})^2 \\
=\ & \Big( \int_0^1 \operatorname{tr} \big( ((\nabla_{\boldsymbol{\alpha}_{\ell+1}} h_{\mathbf{W}_\ell + t\mathbf{U}, j}) \odot \eta'((\mathbf{W}_\ell + t\mathbf{U})\boldsymbol{\alpha}_\ell)) \boldsymbol{\alpha}_\ell^T \mathbf{U}^T \big) dt \Big)^2 \\
=\ & \Big( \int_0^1 \big( (\nabla_{\boldsymbol{\alpha}_{\ell+1}} h_{\mathbf{W}_\ell + t\mathbf{U}, j}) \odot \eta'((\mathbf{W}_\ell + t\mathbf{U})\boldsymbol{\alpha}_\ell) \big)^T \mathbf{U}\boldsymbol{\alpha}_\ell dt \Big)^2 \\
\leq\ & \Big( \int_0^1 ||(\nabla_{\boldsymbol{\alpha}_{\ell+1}} h_{\mathbf{W}_\ell + t\mathbf{U}, j}) \odot \eta'((\mathbf{W}_\ell + t\mathbf{U})\boldsymbol{\alpha}_\ell)||_2 ||\mathbf{U}||_{S^\infty} ||\boldsymbol{\alpha}_\ell||_2 dt \Big)^2 \\
\leq\ & ||\mathbf{U}||_{S^\infty}^2 ||\boldsymbol{\alpha}_\ell||_2^2 \max_{t \in [0,1]} ||(\nabla_{\boldsymbol{\alpha}_{\ell+1}} h_{\mathbf{W}_\ell + t\mathbf{U}, j}) \odot \eta'((\mathbf{W}_\ell + t\mathbf{U})\boldsymbol{\alpha}_\ell)||_2^2.
\end{aligned}$$

The important term here is

$$||(\nabla_{\boldsymbol{\alpha}_{\ell+1}} h_{\mathbf{W}_\ell + t\mathbf{U}, j}) \odot \eta'((\mathbf{W}_\ell + t\mathbf{U})\boldsymbol{\alpha}_\ell)||_2^2.$$

If we approximate this term locally at $t = 0$, then this term can be bound

$$||(\nabla_{\boldsymbol{\alpha}_{\ell+1}} h_{\mathbf{W}_\ell + t\mathbf{U}, j}) \odot \eta'((\mathbf{W}_\ell + t\mathbf{U})\boldsymbol{\alpha}_\ell)||_2^2 \big|_{t=0} \leq ||(\nabla_{\boldsymbol{\alpha}_{\ell+1}} h_{\mathbf{W}_\ell, j})||_2^2 \max_{x'} \frac{d}{dx} \eta(x)|_{x=x'}.$$

Union bound can be applied to make this apply to all data points. This recovers the form in Section 3.2. The second term, $|h_{n,\boldsymbol{\phi},j} - h_{n,\boldsymbol{\theta},j} - \langle \nabla_{\boldsymbol{\theta}} h_{n,\boldsymbol{\theta},j}, \boldsymbol{\phi} - \boldsymbol{\theta} \rangle|$, can be bound for $\mathbf{W}_\ell$ to $\mathbf{W}_\ell + \mathbf{U}$ with

$$\begin{aligned}
&\max_j |h_{\mathbf{W}_\ell + \mathbf{U}, j} - h_{\mathbf{W}_\ell, j} - \langle \nabla_{\mathbf{W}_\ell} h_{\mathbf{W}_\ell, j}, \mathbf{U} \rangle| \\
=\ & \Big| \int_0^1 \big( (\nabla_{\boldsymbol{\alpha}_{\ell+1}} h_{\mathbf{W}_\ell + t\mathbf{U}, j}) \odot \eta'((\mathbf{W}_\ell + t\mathbf{U})\boldsymbol{\alpha}_\ell) - (\nabla_{\boldsymbol{\alpha}_{\ell+1}} h_{\mathbf{W}_\ell, j}) \odot \eta'((\mathbf{W}_\ell)\boldsymbol{\alpha}_\ell) \big)^T \mathbf{U}\boldsymbol{\alpha}_\ell dt \Big| \\
\leq\ & \int_0^1 ||(\nabla_{\boldsymbol{\alpha}_{\ell+1}} h_{\mathbf{W}_\ell + t\mathbf{U}, j}) \odot \eta'((\mathbf{W}_\ell + t\mathbf{U})\boldsymbol{\alpha}_\ell) - (\nabla_{\boldsymbol{\alpha}_{\ell+1}} h_{\mathbf{W}_\ell, j}) \odot \eta'((\mathbf{W}_\ell)\boldsymbol{\alpha}_\ell)||_2 ||\mathbf{U}||_{S^\infty} ||\boldsymbol{\alpha}_\ell||_2 dt \\
=\ & ||\mathbf{U}||_{S^\infty} ||\boldsymbol{\alpha}_\ell||_2 \int_0^1 ||(\nabla_{\boldsymbol{\alpha}_{\ell+1}} h_{\mathbf{W}_\ell + t\mathbf{U}, j}) \odot \eta'((\mathbf{W}_\ell + t\mathbf{U})\boldsymbol{\alpha}_\ell) - (\nabla_{\boldsymbol{\alpha}_{\ell+1}} h_{\mathbf{W}_\ell, j}) \odot \eta'((\mathbf{W}_\ell)\boldsymbol{\alpha}_\ell)||_2 dt.
\end{aligned}$$

The important term here is

$$\int_0^1 ||(\nabla_{\boldsymbol{\alpha}_{\ell+1}} h_{\mathbf{W}_\ell + t\mathbf{U}, j}) \odot \eta'((\mathbf{W}_\ell + t\mathbf{U})\boldsymbol{\alpha}_\ell) - (\nabla_{\boldsymbol{\alpha}_{\ell+1}} h_{\mathbf{W}_\ell, j}) \odot \eta'((\mathbf{W}_\ell)\boldsymbol{\alpha}_\ell)||_2 dt$$

$$\leq \quad \frac{1}{2} \max_t ||\frac{d}{dt}((\nabla_{\boldsymbol{\alpha}_{\ell+1}} h_{\mathbf{W}_\ell + t\mathbf{U}, j}) \odot \eta'((\mathbf{W}_\ell + t\mathbf{U})\boldsymbol{\alpha}_\ell))||_2.$$

It is possible to bounded this term, but it is highly pessimistic. Instead, approximating this by the quantity around $t = 0$ gives

$$\tilde{\leq} \frac{1}{2} (||((\nabla_{\boldsymbol{\alpha}_{\ell+1}} h_{\mathbf{W}_\ell, j}) \odot \eta'((\mathbf{W}_\ell)\boldsymbol{\alpha}_\ell)) \odot \mathbf{U}\boldsymbol{\alpha}_\ell||_2 + ||(\frac{d}{dt}\nabla_{\boldsymbol{\alpha}_{\ell+1}} h_{\mathbf{W}_\ell + t\mathbf{U}, j})|_{t=0} \odot \eta'((\mathbf{W}_\ell)\boldsymbol{\alpha}_\ell))||_2).$$

For deep networks, the second term will vanish faster then the first. It is ignored heuristically. We separate this into

$$\tilde{\leq} \frac{1}{2} ||\nabla_{\boldsymbol{\alpha}_{\ell+1}} h_{\mathbf{W}_\ell, j}||_\infty ||\eta'(\mathbf{W}_\ell \boldsymbol{\alpha}_\ell))||_\infty ||\mathbf{U}\boldsymbol{\alpha}_\ell||_2 \tilde{\leq} \frac{1}{2} ||\nabla_{\boldsymbol{\alpha}_{\ell+1}} h_{\mathbf{W}_\ell, j}||_\infty ||\eta'(\mathbf{W}_\ell \boldsymbol{\alpha}_\ell))||_\infty ||\mathbf{U}||_{S^\infty} ||\boldsymbol{\alpha}_\ell||_2.$$

Combining with (16) recovers the term in the text,

$$\leq \frac{1}{2} ||\mathbf{U}||_{S^\infty}|^2 ||\nabla_{\boldsymbol{\alpha}_{\ell+1}} h_{\mathbf{W}_\ell, j}||_\infty ||\eta'(\mathbf{W}_\ell \boldsymbol{\alpha}_\ell))||_\infty |\boldsymbol{\alpha}_\ell||_2^2.$$