[Reviews · NeurIPS 2015]

Submitted by Assigned_Reviewer_1

Quasi-Newton refers to a very specific kind of 2nd-order method based on low-rank updates to an approximate Hessian

Isn't the Schatten-inf norm just the spectral norm/matrix 2-norm?

Line 68: D is a square matrix as in eqn. 1, right?

How are you point-wise multiplying it against a matrix the size of the weight matrix?

Also, this doesn't correspond to diagonal preconditioning even if it were valid as far as I can tell. Why is it the square root?

Why is there an inverse?

I have no idea what's going on here.

Line 128: Evidence of this?

Line 286: What do these bounds have to do with the algorithm described in Algorithm 2?
Summary: This paper extends some prior [1] work done on using local bounds based on spectral norms to derive an iterative optimization strategy (where the bounds are minimized at each step).

This line of research is interesting, but this paper isn't particularly well written and seems rushed.

I'm also not convinced that the experiments on neural networks were fairly done.

Ideally, there should be evidence that the competing methods were well calibrated, and the example problems should be

ones from other papers looking at optimization speed (instead of being tried for the first time in this paper).

Submitted by Assigned_Reviewer_2

I find the explanation of the ADAspectral and RMSspectral interesting. A few comments that I have are:

(1) Readability for wider audience:

- It might be useful (maybe in the supplementary material) to provide a

more verbose description of how one gets the update rule for either of

the methods. Or even better, code ! Most readers will first want to know

how hard it is to implement and how well it works before trying to

understand the math. From this perspective, algorithm 1 looks cryptic

with the #-operator and b-operator in there.

- I know it is said, but make it more clearer from the beginning that these

approach amounts to having a different update rule for the weights

(which are matrices) compared to biases. And that for the weights it

involves an SVD (or approximate SVD) over the gradient of these weights.

I think the paper will have a much bigger impact if it is really easy to

follow and clear from the start what the end result will be, even for one

that is not used to these things.

(2) Intuitively one way of understanding this algorithm is the following

(please correct me if I'm wrong). Say you set D to I (identity). Than you

are simply using the sign of the gradient for the biases (multiply this by

the norm of the gradient), and projecting

the gradients of the weights on a sphere of radius equal to the largest

singular value. Now if D is different from I, it is almost like you rely on

D completely to tell how much more you should move along an axis vs

another. Again for the biases this is simpler to see than for the weights.

* First of all, it would be interesting to compare this with something like

Rprop. Rprop basically keeps only the sign of the gradient and ignores its

magnitude. The RMSspectral seems to project the gradients on some sphere

like Rprop (just that you do a rescaling of the gradients with D before

that and the radius of the sphere is not the learning rate, but the norm of

the gradient), but

then to change the shape of the sphere based on D.

* Secondly it would be interesting to know how vital is for D to be a good

approximation of the curvature. E.g. it feels that if I make D to be I,

I would get an algorithm that probably is much slower than SGD !? Is this

because now I don't have a tight bound on F(y). Is there some underlying

assumption on D in the derivation of the algorithm that can be easily

made visible (e.g. the bound becomes tighter if D becomes closer to the

curvature?).

(3) I find it hard to compare the results the authors have with other results

in the literature because they present the results in terms of normalized

time (instead of actual time). It would have been useful to run the same

experiment as the one of some other already published paper on

optimization and present the results in a similar fashion (e.g. the deep

autoencoder on MNIST or something else).

Minor things:

* tiny spelling mistakes (e.g. line 403: epcohs)

* figure 3, what does normalized time mean?
Summary: The paper introduces two new optimization methods (based on RMSprop and adagrad) that seem to deal much better with the non-euclidean structure of the error surface.

Submitted by Assigned_Reviewer_3

The paper is an extension to a previously published method, "spectral SGD", in which for each update to a weight matrix, an SVD is done and the singular values are replaced with some fixed value.

The basic extension is to do this after an initial diagonal preconditioning similar to what is done in Adagrad.

While the method is interesting, and the results seem promising, there is a flaw in spectral SGD.

The explanation the authors give for spectral SGD is very fancy-sounding, with lots of discussion of spaces and metrics, but I don't think they have considered it carefully from the perspective of convergence.

A necessary but not sufficient condition for convergence of any variant of SGD is that at an optimal value of the parameters (i.e. at the objective max/min),

the expected parameter change should be zero.

But this is not the case in SSGD.

Assume there is just one parameter, and at the optimal value, the parameter derivative is -1.0 99% of the time, and +99.0 1% of the time.

The average parameter derivative is obviously zero.

But assuming the minibatch size is fairly small (e.g. less than 100), the derivative per minibatch will be negative more often than it is positive.

Since SSGD only takes the sign of the derivative in this 1-d case, you'll get an update that's negative on average.

So no convergence.

I believe the reason this works is that it it's approximating a natural gradient type of update, and I think it would be better to formulate it in that way.

- Reducing my score slightly after the author rebuttal. The authors say: 'Regarding the "flaw": recall that the #- and b-operators do keep the magnitude of the gradients. Hence, the expected gradient is zero and the sharp operator does not alter this. Hence, the reviewer's example does not hold. ' I did not notice that they kept the magnitude of the gradients; however, it only takes a slight modification of my example to a two-dimensional example (with the other dimension behaving any way you choose) to generate a counterexample.

This is so obvious that I'm surprised the authors couldn't see it. The authors also said: 'The similarity to RPROP is not obvious to us. For instance, unlike RPROP the #-operator can change the sign of a coordinate during the non-linear transformation. '.

I guess this relates to a different reviewer's comments.
Summary: A reasonable paper, but incremental contribution is not that huge, and there is a flaw in the basic idea of the prior paper.

Submitted by Assigned_Reviewer_4

The claim/implication that this approach is particulary well-suited for deep learning per se seems a bit of a stretch and a blatant appeal to the recent interest in "deep" models -- the justification seems to rely on the use of softmax top layers, which many other non-deep models use as well.

It is good that author's properly seem to account for the SVD costs in their timings, and that their finding that fast popular randomized projection style SVDs suffice seems plausible.

Figure 3 results seem promising, including 5x speedup for RMSspectral.

Most disappointing is that the authors do not discuss any limitations or tradeoffs.

For example, why did they not report AdaGradSpectral results for CIFAR 3-layer CNN in Figure3(c)?

Did it perform worse than non-spectral?

If so, why?

In general, when will their approach do better or worse than non-spectral methods?

Presumably a flat eigen spectrum would make SVD approximations work well, so that could be one issue.

Lack of admitting/discussing these issues makes this paper much less impactful, and much less a clear accept, that it otherwise could be.
Summary: Proposes a novel preconditioning for SGD and offers reasonable experimental work to indicate its promise in speeding up training times.

Submitted by Assigned_Reviewer_5

This paper can be seen as a direct extension of [1]. It combines the idea presented in [1] to use a non-Euclidean norm to derive a better first-order approximation (majorization) of the objective function of an RBM with variable-metric approximations which lead to preconditioned first-order methods. In combination, one obtains a proximal term in the objective function approximation which uses a elementwise-scaled Schatten-infty norm. As preconditioners the dynamic preconditioners of ADAgrad and RMSprop are used.

The paper is well-written and the it is described in a clear way how to implement the new algorithms.

Although the idea of using a non-Euclidean norm was already presented in [1], the experiments clearly show the improvement obtained by combining this idea with preconditioning.

These results as well as many other papers indicate that it is often beneficial to use "curvature" information (quasi-Newton methods, truncated Newton (aka Hessian-free) methods, natural gradient methods,...). Consequently, the question arises whether the algorithms presented here can also benefit from using second-order information explicitly and how they compares to existing methods which do so.

Note: * Typos in title of Section 2 and l. 302.
Summary: This paper combines the idea of [1] to use a non-Euclidean norm to approximate the objective function of ML training of an RBM with preconditioning. This yields new algorithms which seem to outperform the related ADAgrad and RMSprop methods as well as the algorithm originally proposed in [1].

Submitted by Assigned_Reviewer_6

Iterations of (stochastic) gradient descent can be written as the minimization of a linearization of the objective function with a squared Euclidean norm on the distance between the parameters of the current and next iteration. Based on earlier results in training restricted Boltzmann machines (RBM), the authors consider non-Euclidean, in particular the Schatten-infinity norm, combined with preconditioning matrices D as given by the RMSprop and ADAgrad algorithms. It is shown that resulting subproblems can be solved exactly by applying the sharp-operator, and approximately, but more efficiently using low-rank approximations. The approach is extended for use in neural networks by using approximate upper bounds instead of tight upper bounds. The algorithm is applied to both both RBM and neural network training and shown to compare favorably with competing methods.

Some minor comments: - p.1, abstract "derive a novel ... algorithms", remove either "a" or "-s" - p.2, Section 2 heading "Non-Eucldiean" - p.3, (see also A.1) definition of S# = arg min_x {(s,x) - 1/2||x||^2}

the minus sign seems incorrect, note that choosing x = -alpha * s allows

us to decrease the objective to minus infinity by increasing alpha. - p.5, rectified linear units do not have Lipschitz continous gradients as

required by (6). Nevertheless on page 7 the experiments use ReLU. Some

comments on this would be useful. - p.7, "sweeps between per iteration", "between per" - p.7, "to macth [23]", "match" - p.7, "multiple nat improvement", "net"? - p.8, "only 40 epcohs", "epochs" - p.8, "To further demonstarte", "demonstrate" - p.8, "While they eventually achieve similar accuracy rates, RMSspectral reaches that rate five times faster", actually RMSprop gives slightly better results after iteration 350, also it is about 3.5 times faster, not 5. - p.8, The Cifar-10, 3-layer CNN is trained using only two methods, how does this compare with the methods used in the other plots of Figure 3?
Summary: The proposed algorithm based on a combination of non-Euclidean norms with the preconditioners given by RMSprop and ADAgrad are shown to be computationally efficient compared to several competing methods.

Author Feedback
Author rebuttal: We want to thank all the reviewers for their helpful comments. We will address minor issues in the manuscript, and directly respond to reviewers below.

R#1
We are in fact now ready to release our code on github both for torch and separately for matlab using the matconvnet toolkit (based off of the torch implementation).
The SSD algorithm corresponds to D=I and already outperforms SGD with a notable improvement when D is updated. We we add a note to the paper to make the relationship that SSD corresponds to D=I as well, so this relationship is clearer.
The concept of curvature does not immediately translate into the spectral space. We will add some intuition in our revision on this as well as additional explanations on the #- and b-operators for completeness.
We use normalized time so that the relationship to the number of iterations is cleaner. We will note the conversion to actual time in the paper, which is .0082 seconds using a Tesla K40c GPU.

R#2:
Regarding the "flaw": recall that the #- and b-operators do keep the magnitude of the gradients. Hence, the expected gradient is zero and the sharp operator does not alter this. Hence, the reviewer's example does not hold.
The similarity to RPROP is not obvious to us. For instance, unlike RPROP the #-operator can change the sign of a coordinate during the non-linear transformation.

R#3:
The RELU comment is valid. However, in such a case, we can continue operating with the subgradient. We will mention this in the revision.
The Cifar-10 3-layer CNN holdout set is shown with two algorithms for clarity, because RMSspectral and RMSprop outperform all others. However, for completeness, we can add the performance all algorithms to this figure.
We will fix the typo in the equation.

R#4:
We will clarify that the Schatten-inf norm is alternatively known as the spectral norm or the matrix 2-norm.
Here is an example for 128: log-sum-exp functions is a known smoothing technique for ell_inf norm (e.g., Nesterov's proximity smoothing in the dual).
Line 286: these bounds allow us to derive gradient descent step-sizes in a deterministic sense.
R#5:
We will clarify why we use the element-wise product form instead of a standard matrix multiplication.
R#6:
In regression tasks, we found this method to be slightly slower in iteration. Given that per iteration is costlier, it may be better to directly use SGD or its enhanced versions.